# Effects of stimulus emotional content on gaze pattern: An eye-tracking study

Andrés Castellanos-Chacón🆔*, Daniela Arias-Otero, Valeria Uribe-Jaramillo, Juan David Leongómez🆔◎, Milena Vásquez-Amézquita🆔◎*

Faculty of Psychology, Universidad El Bosque, Bogota, District of Columbia, Colombia

◎ These authors contributed equally to this work.
* gcastellanos@unbosque.edu.co (AC-C); mvasquezam@unbosque.edu.co (yMV-A)

## Abstract

The attentional system tends to prioritize negative stimuli in the early stages of processing, favoring threat detection. However, it is unclear whether this bias is maintained or reversed toward positive stimuli at later stages. In this study, we used a free-viewing paradigm with eye tracking to examine early and late attentional biases toward negative, positive, and neutral stimuli (humans in emotionally unloaded activities) versus control stimuli (inanimate objects) in 122 participants without affective disorders (64 men, 58 women). We fitted generalized linear mixed models with random intercepts for stimuli and random intercepts and slopes for participants, and used non-parametric bootstrap resampling to obtain robust estimates and confidence intervals. Additionally, the number of first fixations was analyzed with a COM-Poisson. Results showed that participants fixated faster ($\chi^2(3) = 97.55$, $p < .001$) and for longer durations ($\chi^2(3) = 337.45$, $p < .001$) on negative stimuli compared to the other categories, confirming a negativity bias in early attention In late attention, we found longer total fixation durations ($\chi^2(3) = 200.24$, $p < .001$) and a greater number of fixations ($\chi^2(3) = 207.02$, $p < .001$) for negative stimuli, contradicting the hypothesis of a positivity shift during emotional regulation. This sustained negativity bias may reflect an adaptive regulatory process in which individuals allocate attentional resources to threat-related information to enhance learning and emotional preparedness. Future studies should examine these effects across diverse sociocultural settings and in clinical populations.

## Introduction

Attention involves a series of processes facilitating the selection of relevant information from the context [1,2]. It includes alerting mechanisms [3] and the ability to either voluntarily attend to or ignore presented stimuli [4], allowing the transformation of sensory input into information processed in working memory [5]. However, our brain can attend to stimuli or ignore them, depending on the load assigned to a stimulus, and

provided the original author and source are credited.

**Data availability statement:** All files are available from the study in https://osf.io/c89nx/.

**Funding:** Authors: Castellanos-Chacón, A., Arias-Otero, D., Uribe-Jaramilo, V., Vásquez-Amézquita, M. y Leongómez, J. Grant numbers awarded to each author: 1 The full name of each funder: Ministerio de Ciencia Tecnología e Innovación URL of each funder website: https://minciencias.gov.co/viceministerios/talento/vocaciones/jovenes Sponsors had no role in the study.

**Competing interests:** The authors have declared that no competing interests exist.

this is known as bias, which in a general sense, refers to a systematic deviation in the processing of the information we attend to and which leads us to have a preference for certain stimuli over others [6]. Attentional biases, a specific type of bias, are systematic tendencies to allocate attention preferentially to certain types of stimuli, which influences the way in which individuals interpret and respond to their environment [7]. At any given time, the human sensory system is exposed to an overwhelming number of stimuli. This bias involves processing certain information more readily or intensely than other available inputs [8]. Specifically, threat-related attentional bias describes the preference for processing threat-related stimuli over positive or neutral stimuli [9].

Emotions enable the evaluation of decisions and the execution or inhibition of actions [10,11], prompting changes in an individual's behavior based on the environment. Additionally, emotions function as modulators of higher cognitive processes such as attention, memory [12] and decision-making [13]. Emotionally salient stimuli tend to capture attention more effectively than neutral stimuli, as they are often linked to adaptive responses related to survival, threat avoidance, and reward-seeking behaviors [10,14]. In addition, inhibitory processes play a crucial role in the regulation of attention, as they allow individuals to suppress irrelevant or distracting stimuli in favor of more relevant information [10]. This selective filtering mechanism ensures that cognitive resources are allocated efficiently, especially in environments with competing stimuli.

The literature shows a consensus that emotional stimuli capture attention faster [15–19] and for longer durations [18,19] than stimuli without emotional content. However, there is an ongoing debate about the directionality of this bias: whether negative stimuli elicit stronger attentional capture due to their threat-related significance ("negativity bias") or whether positive stimuli draw attention preferentially in later stages of processing ("positivity bias") [17,18].

### Early attention bias: Positivity vs. Negativity

Some studies suggest that negative stimuli are prioritized during early visual processing, which aligns with evolutionary theories emphasizing the necessity of detecting threats quickly [20,21]. This bias is thought to reflect an adaptive response, where missing a threat is costlier than missing a reward [22]. Experimental evidence using eye-tracking supports this claim, demonstrating that time to first fixation (TFF) and first fixation count (FFC) tend to be lower for negative stimuli compared to positive or neutral stimuli [16].

However, alternative findings indicate a preference for positive stimuli in early processing stages [23,24]. This could suggest an avoidance mechanism for negative stimuli or a broader cognitive strategy that favors positive reinforcement [25]. Another possibility is that methodological differences, such as whether emotional and neutral stimuli compete simultaneously, influence results [26].

### Late attention bias: Positivity vs. Negativity

While early attention biases have been widely studied, findings regarding late attention allocation are less consistent. Some research suggests that after an initial

negativity bias, attention may shift towards positive stimuli in later stages [27,28], possibly as a recovery mechanism from negative arousal [29]. However, other studies find sustained attention to negative stimuli, suggesting prolonged engagement rather than disengagement [30,31].

These discrepancies may arise from differences in methodologies. Paradigms using free-viewing eye-tracking tasks provide a more ecologically valid approach to studying attentional biases, as they capture natural gaze behavior over time [16,19]. Unlike reaction-time-based tasks, which measure isolated moments of processing, eye-tracking enables the distinction between early and late attention by analyzing fixation duration and gaze patterns [32].

## Eye tracking: Measurement of the temporal course of attention

Eye-tracking is a widely used method for studying attentional biases toward emotional stimuli, as it allows precise measurement of visual attention allocation at different processing stages [19,33–35]. This technique differentiates between early attention, assessed via time to first fixation (TFF), first fixation duration (FFD), and first fixation count (FFC) [32,36], and late attention, measured by total fixation duration (TDF) and total number of fixations (TNF) [13,37,38].

Early studies demonstrated that emotional stimuli both positive and negative capture attention more quickly than neutral stimuli [16,19]. However, subsequent research indicated that negative stimuli dominate early attention [39], whereas positive stimuli generate greater late attention [27,29,31].

Despite these findings, previous studies often lacked direct competition between positive and negative stimuli or did not control for confounding factors such as the presence of humans in neutral images [31, 40]. Addressing these methodological limitations, by concurrently presenting positive, negative, and neutral stimuli while controlling for human-related content in neutral images, can provide a more accurate and ecologically valid assessment of natural attentional biases in non-clinical populations.

## The currently study

Despite substantial research on attentional biases, few studies have simultaneously examined early and late attention allocation toward positive and negative stimuli while controlling for neutral human-related content and inanimate control stimuli. Additionally, previous research often focuses on clinical or subclinical populations (e.g., individuals with anxiety or depression), limiting generalizability about natural patterns in the general population [41,42].

This study aims to fill this gap by investigating how individuals without affective disorders allocate attention toward negative, positive, and neutral stimuli in a free-viewing eye-tracking paradigm. By including non-emotional human-related stimuli and control objects, this study accounts for the potential confound of social relevance in attentional allocation. Furthermore, we examine gender differences in attentional biases, given prior evidence suggesting potential variations in emotion processing [43]. Based on the background, we expected individuals, regardless of gender, to show: 1) increased early and late attention to emotional stimuli (negative or positive) than to non-emotional ones neutral or control; 2) increased early attention to negative stimuli, followed by positive, neutral, and control stimuli, respectively, reflecting an early negativity bias towards threat; 3) increased late attention to positive stimuli, followed by negative, neutral, and control stimuli, indicating a positivity bias for emotional regulation or stress avoidance; and 4) increased early and late biases toward non-emotional neutral stimuli with human presence compared to inanimate controls.

Validating these hypotheses could help accurately identify externally influenced attentional biases related to emotional stimulus content. This approach could then serve as a parsimonious and valid design for evaluating attentional biases induced by internal states or individual characteristics, such as anxiety or depression levels, which have been of great interest in recent years [44–47].

## Materials and methods

### Participants

We recruited 156 Colombian participants living in the city of Bogota or neighboring areas, including people of any socio-economic level who could read and write. The final sample consisted of 122 participants (58 women; 64 men) between the ages of 18 and 35 (M±SD = 22.04 ± 3.6; women: 21.48 ± 3.56; men: 22.56 ± 3.67).

Participants outside the age range of 18–35 were excluded, along with those clinically diagnosed with mental or neurological disorders and smokers (the latter criterion being part of a larger study measuring heart rate frequency). Out of a total of 156 participants, 34 were excluded because they scored above 19 on the Beck Depression Scale, indicating moderate depressive symptoms [48,49].

Although the sample size was not specifically determined based on statistical power, it was decided to increase it compared to prior studies examining the influenced attentional biases related to emotional stimulus content (e.g., $n = 24$, Calvo & Lang [16]; $n = 40$, Fernández-Martín et al. [18]; $n = 54$, Lea et al. [50]; $n$ Exp 1 = 23 & $n$ Exp 2 = 32, Nummenmaa et al. [19]; $n = 77$, Sears et al. [51]). Given the within-subject design and repeated-measures structure of the eye-tracking data, we expect high sensitivity for detecting attentional biases. However, we acknowledge the limitation of our study and the importance of power analyses in study planning, and encourage future research to incorporate a priori power calculations to further strengthen the robustness of the findings.

### Equipment and materials

**Eye tracker.** The Tobii Pro Fusion Eye Tracker, synchronized with Tobii Pro Lab Software (Version 1.162.32461 x64), provided ocular registration. It offers a spatial visual angle resolution of less than 0.2° and a temporal resolution of 250 Hz. The software ran on a Lenovo Thinkcentre M83 PC, with an Intel Core i7-4790, 3.60 Hz processor, and 16 GB RAM. Stimuli were presented on a 21.5-inch Lenovo ThinkVision E2223s monitor.

**Questionnaire.** One sociodemographic and two psychological questionnaires were administered using Qualtrics and Google Forms. The former included questions about age, gender, education level, and neurological or psychological/psychiatric conditions to filter participants. The Beck Depression Inventory (BDI) and State-Trait Anxiety Inventory, Trait version (STAI-T), were used to assess levels of trait depression and anxiety. Participants with BDI scores outside the range of 0–19, or STAI-T scores above 18, were excluded [48,49,52,53].

**Stimuli and experimental paradigm.** Initially, 71 stimuli were preselected from the Nencki Affective Picture System (NAPS) [54] and Open Affective Standardized Image Set, OASIS [55] image banks and validated with a population similar to the one evaluated (228 participants). Arousal and valence were evaluated for the stimuli, and those that maintained the evaluation in both dimensions were used for the experiment.

Subsequently, 16 stimuli were selected from each category (64 in total). For more details, see Stimuli and experimental paradigm in the S1 File. For the construction of the 32 trials with the four competing stimuli, 16 trials were first constructed with the 64 images obtained, each of which contained one randomly selected image of each of the four types of stimuli (Positive, negative, control, and neutral).

Subsequently, another 16 trials were conducted with the same 64 stimuli in the same manner, but without allowing images that were presented together in the first 16 created stimuli to appear together again. To ensure that the stimuli were not in the same disposition, the location of the four stimuli types was equally likely to appear in each corner (Fig 1):

In Tobii Pro Lab software, a block was created where 32 stimuli were presented for 8 seconds. Each stimulus consisted of four simultaneous images: a negative photograph depicting harm, weapons, death, or poverty; a positive photograph featuring animals, enjoyable activities, or sports; a neutral photograph showing people engaged in non-emotional activities like typing or sitting; and a control photograph displaying inanimate objects without emotional content, such as a bed or a bottle. Each image measured 453 pixels in height and was located in one of the four corners of the screen.

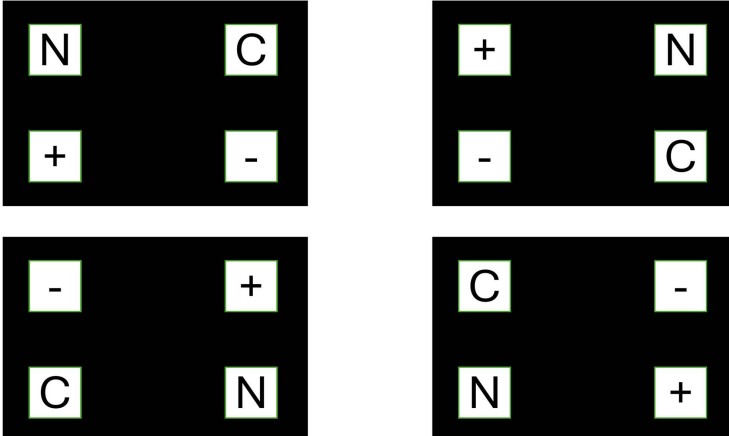

**Fig 1. Presentation of experimental paradigm stimuli.**

Each photograph with emotional content appeared twice during the experiment alongside three other stimuli in each presentation. Stimuli were placed 680 pixels apart, with content counterbalanced across the four corners of the screen, so that each stimulus was placed in each corner eight times. Participants viewed a black screen with a central cross for 500 ms before each stimulus. They were given the following the instruction: "look at the stimulus freely as if you were looking at a photo album, the only condition is that you must look openly at the images once they appear on the screen", ensuring open attention and preventing data loss due fixation on the cross.

**Subjective evaluation of stimuli.** In Qualtrics, all stimuli in the experimental paradigm were individually and randomly designed and presented for subjective evaluation of emotional content with eye tracking. Participants first rated the stimuli on valence, indicating positivity or negativity on a scale from +4 (very pleasant) to −4 (very unpleasant). Then, they rated the stimuli on arousal, indicating emotional intensity on a scale from 0 (not at all activating) to 9 (very activating).

## Procedure

While this study was not pre-registered, all materials, data, and code are openly available for verification (see S1 File). Data were collected between November 11, 2022 and November 30, 2023, and analyzed in 2024.

The study began with participants receiving questionnaires to verify inclusion and exclusion criteria one day before the experimental session. Participants were presented with an online consent form to agree to data collection. If they agreed, they proceeded to answer the questionnaires. After confirming participants met the inclusion criteria, they were individually allotted to the Experimental Psychology Lab of Universidad El Bosque the following day.

In the laboratory, participants received initial covert written informed consent, followed by full consent. Initially, participants were not explicitly told about the collection of their eye movement data to prevent observational bias (Hawthorne Effect) [56,57]. They were only informed about viewing images with emotional content. After signing the consent form, participants started the experimental protocol.

First, participants were positioned between 60 and 70 cm from the screen. Eye tracker calibration consisted of following 9 points on the screen until participants achieved an accuracy index of less than 0.5°. Once calibrated, the experimental paradigm began.

Participants were first instructed to focus on the cross before advancing to the photograph and then freely visualize the stimuli. Then, participant underwent a training phase resembling the actual experimental paradigm, involving the presentation of two stimuli. Following this, the experimental trials were presented in different random order to participants. Once

finished, participants signed the full informed consent, containing the research's overarching objective and the opportunity to accept or reject the use of their eye-tracking data for the study.

## Statistical analysis

Generalized linear mixed models (GLMMs) were fitted for each dependent variable. Time-based variables (TFF, FFD, TDF, TNF) were modeled with a Gamma distribution and a log link, while the First Fixation Count (FFC) was analyzed using a count-based GLMM with a Conway–Maxwell–Poisson (COM-Poisson) distribution, which flexibly accounts for under- or over-dispersion in count data. All models included random intercepts for participants and stimuli, and by-participant random slopes for stimulus content.

For early attention, the modeled variables were: Time to First Fixation (TFF), First Fixation Duration (FFD), and FFC. For late attention, the modeled variables were: Total Duration of Fixation (TDF) and Total Number of Fixations (TNF). The main fixed effects included stimulus content and participant gender, while the random effects included intercepts for participant and stimulus ID, plus random slopes for stimulus content within participant.

To obtain robust estimates and assess the stability of effects, non-parametric bootstrap resampling with 1000 iterations was used to compute empirical confidence intervals (CIs) for the fixed effects. Statistical inference was based on whether these CIs included zero: effects whose 95% CI bounds did not cross zero were interpreted as statistically significant. In all main results, the bootstrap-based conclusions matched those obtained using conventional methods.

All statistical analyses were performed using R version 4.2.2, and all code is provided in the S1 File https://doi.org/10.17605/OSF.IO/C89NX

## Transparency and openness

The "participants" section of the article includes studies considered for the sample size. Detailed information about the stimuli used in the experimental paradigm is provided in Stimuli and experimental paradigm within the S1 File. Additionally, data analysis of the S1 File, includes all code used for the analysis, offering a comprehensive guide for replicating and verifying our findings. No data were omitted, except for the participants who did not meet the inclusion criteria of the experimental paradigm, as mentioned earlier.

This study was not pre-registered due to a lack of information on pre-registration requirements at the time the study was initiated, and because it was not a standard procedure within our research framework when the study was designed. Despite this, we have made efforts to guarantee transparency by providing detailed methods and reproducible analyses, including the sample size justification, stimulus selection, and the complete analysis code along with the data in the S1 File. Furthermore, all hypotheses and statistical models were specified to the ethics committee prior to data collection, minimizing potential bias in the analysis.

## Ethical considerations

The study protocol was approved by the institutional ethics committee of Universidad El Bosque (PIS 002–2020). A first informed consent form was submitted virtually, in order to obtain preliminary data that would allow us to discard those participants who did not meet the inclusion criteria of the research. Subsequently, participants who met the criteria reported to the laboratory the following day. In the laboratory, participants signed a covert written consent form, in which participants were not initially informed that their eye movements were being recorded. Personal data were treated confidentially and anonymously to ensure privacy and compliance with data protection regulations. Only aggregated data were used in the analyses and no personally identifiable information was retained. Once the experimental paradigm was completed, full informed consent was provided, explaining the experiment in detail and, in the event of not wishing to participate, ensuring the deletion of all data collected.

We recognize that the use of covert consent procedures poses significant ethical challenges. However, in the context of this study, this approach was deemed necessary to ensure the scientific validity of the results while minimizing any potential risk to participants. Throughout the process, specific measures were taken to safeguard the autonomy of the participants and to respect ethical standards.

In order not to disrupt natural patterns of visual scanning, participants were not initially informed that their eye movements were being recorded. This temporary omission was approved by the institutional ethics committee of Universidad El Bosque (approval number PIS 002–2020), based on the following considerations: (1) the risk to participants was minimal, as only gaze patterns were collected in response to emotionally varied but noninvasive visual stimuli, and no personal biometric data beyond gaze location were recorded; (2) a thorough debriefing and full informed consent was conducted immediately after the task, during which participants were given the opportunity to withdraw their data without penalty; and (3) early disclosure of the specific study objectives would likely have compromised the validity of the data by altering participants' naturalistic visual behavior.

This approach aligns with recent methodological perspectives that support the ethical use of covert eye-tracking procedures under controlled conditions, particularly when transparency may compromise the authenticity of participants' responses. For example, Riege et al. [58] argue that covert eye-tracking can be ethically justified when accompanied by robust debriefing protocols and voluntary data withdrawal options, as it allows researchers to investigate cognitive and attentional processes without introducing reactivity or demand characteristics.

In applying this design, we attempted to balance the need to reduce experimental bias with the ethical imperative to respect participants' rights. No participants expressed discomfort or chose to withdraw after receiving full information about the study, suggesting that the measures taken were effective in maintaining both scientific integrity and ethical responsibility.

## Results

### Early attention

Models predicting Time to First Fixation (TFF), First Fixation Duration (FFD), and First Fixation Count (FFC) are summarized in Table 1. While gender did not have a significant effect on any model, stimulus content had a significant main effect on TFF, FFD and FFC. The interaction between gender and stimulus content was only significant for FFC.

The marginal $R^2$ values were relatively low, indicating that the fixed effects explain a modest proportion of the variance. Random effects captured meaningful participant-level variability. For TFF (Model 1, see details in Table S5 in the S1 File) and FFD (Model 2, see details in Table S13 in the S1 File), random intercepts and slopes for stimulus content showed moderate variance (SD = 0.13–0.36) and strong negative correlations between intercepts and slopes (r = −0.66 to −0.84),

**Table 1. Overall results early attention.**

| Early attention | TFF | | FFD | | FFC | |
|---|---|---|---|---|---|---|
| | $\chi^2$ | p | $\chi^2$ | p | $\chi^2$ | p |
| Intercept | 59.47 | **<.001** | 3266.08 | **<.001** | 6043.41 | **<.001** |
| Gender | .091 | .34 | 1.73 | .19 | 2.13 | .14 |
| Stimulus content | 97.55 | **<.001** | 67.88 | **<.001** | 158.86 | **<.001** |
| Stimulus content * gender | 1.75 | .62 | 3.01 | .39 | 9.03 | **.03** |

*Note*. Effects of Stimulus Content (Negative, Positive, Neutral, and Control) and Participant Gender (Women, Men) on Time to First Fixation (TFF), First Fixation Duration (FFD), and First Fixation Count (FFC). TFF and FFD results are from generalized linear mixed models (GLMMs) with likelihood ratio tests ($\chi^2$), and FFC results are from a COM-Poisson GLMM (Type III Wald $\chi^2$ tests). TFF model: conditional $R^2$ = .169, marginal $R^2$ = .043. FFD model: conditional $R^2$ = .16, marginal $R^2$ = .022. FFC model: conditional $R^2$ = − (could not be estimated due to negligible variance of the random effects), marginal $R^2$ = .094. Significant effects are in **bold**.

suggesting that participants with faster fixation tendencies exhibited greater differentiation between stimulus types. In contrast, the FFC (Model 3, see details in Table S20 in the S1 File) model was singular, with negligible random-effect variance, indicating limited participant-specific variability for this measure.

Given the significant effect of stimulus content on all models, *post-hoc* contrasts were performed (see details in Tables S7, S15, S21 and S22 in the S1 File). These revealed that the negative stimuli was, on average, the first to be observed, followed by the positive stimuli, then the neutral stimuli and finally the control stimuli (Fig 2a). For FFD, we found significant differences between negative and control, positive and control, and neutral and control stimuli (Fig 2b), indicating that it took significantly longer for participants to fixate on the controls compared to other stimuli contents. Finally, considering that for FFC both a main effect of stimulus content and a significant interaction between stimulus content and gender, *post-hoc* tests were conducted for all pairwise comparisons between all combinations of stimuli content and gender. These showed that both men and women fixated first on negative stimuli the most, followed by positive, neutral, and control stimuli (Fig 2c and 2d). All contrasts between contents, with the exception of positive versus neutral stimuli, were significant in both men and women.

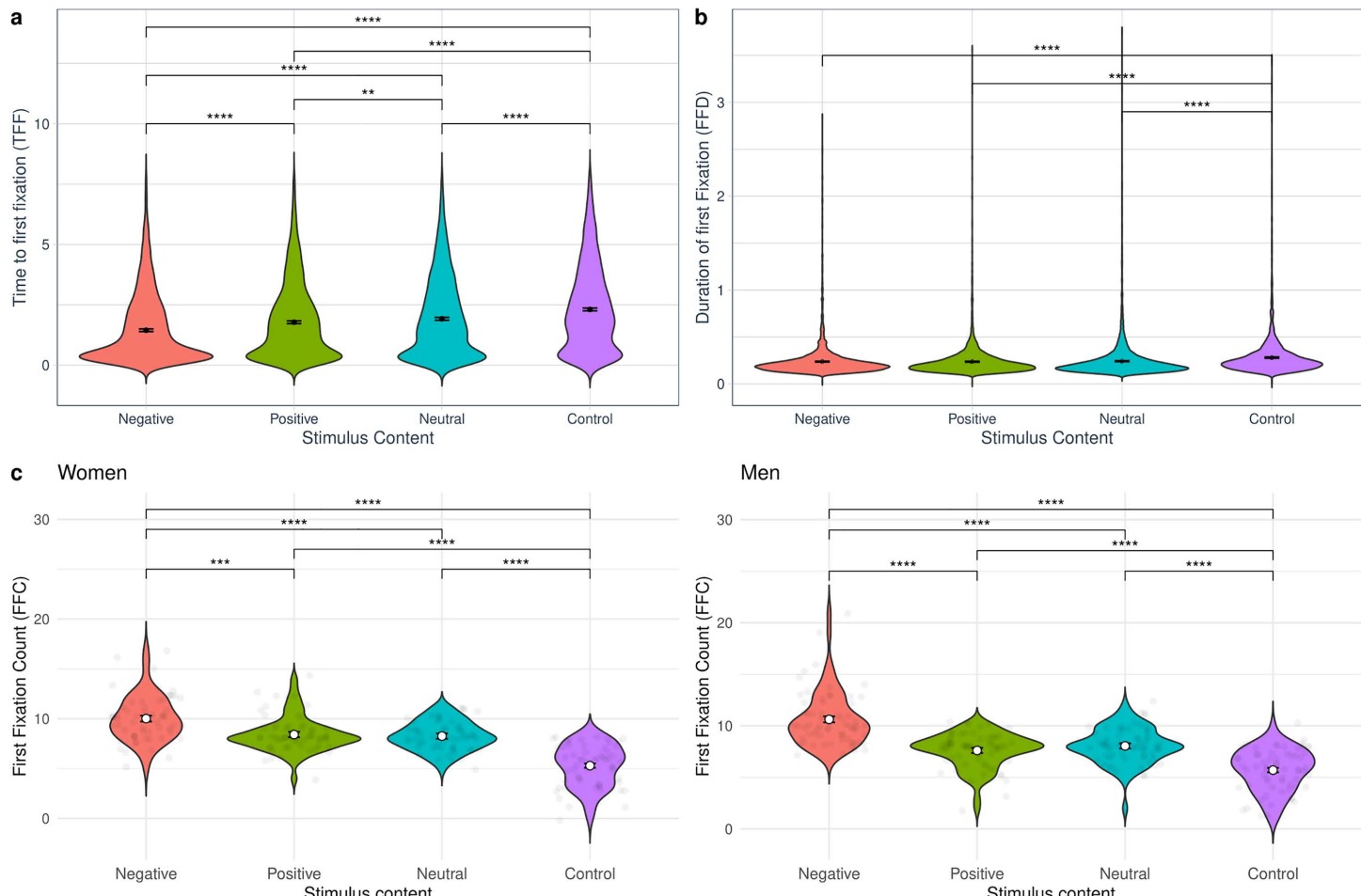

**Fig 2. Early attention.** Effects of stimuli's emotional content (negative, positive, neutral, control) on Time to First Fixation (TFF), First Fixation Duration (FFD), and First Fixation Count (FFC). Considering that for FFC the interaction of gender with stimulus content was significant (Table 1), separate panels by gender are presented. **(a)** TFF; **(b)** FFD; **(c)** FFC of women and FFC of men. Significant contrasts are represented with lines and stars: *p<.05, **p<.01, ***p<.001, ****p<.0001.

To further clarify the interaction between gender and stimulus content, we examined pairwise comparisons within and across stimulus categories. No significant gender differences were observed when comparing men and women within the same stimulus type (e.g., men vs. women for negative stimuli). However, the interaction effect emerged when comparing different combinations of gender and stimulus type (for example, comparing fixations on negative stimuli in women to fixations on positive stimuli in men). This indicates that gender differences in attentional allocation are context-dependent and shaped by the emotional content of the stimuli rather than by gender alone.

### Late attention

Total Duration of Fixations (TDF) and Total Number of Fixations (TNF) models are reported in Table 2. Gender did not have a significant main effect in either model. In contrast, stimulus content had a significant main effect in both TDF and TNF. The interaction between stimulus content and gender was not significant in either case.

As in early attention, the marginal $R^2$ is low indicating that, although the independent variables explain little variance in response, the effects found are consistent and significant. Thus, it is suggested that there may be other potentially individual or contextual factors influencing attentional patterns.

Random effects revealed meaningful individual variability. In the TDF model (Model 4, see details in Table S28 in the S1 File), variability was highest for the random slopes of positive and control stimuli ($SD = 0.28$ and $0.26$), with moderate intercept variability ($SD = 0.25$). In the TNF (Model 5, see details in Table S35 in the S1 File), a similar pattern emerged ($SDs = 0.21$ and $0.20$ for positive and control stimuli; intercept $SD = 0.23$). Stimulus-level variance was negligible in both models.

Pairwise contrasts between stimuli content categories for each gender (see details in Table S29 for model 4, S36 for model 5 in the S1 File), revealed that both men and women fixated the longest and the most times on negative stimuli, followed by positive, neutral and control stimuli, in that order (Fig 3).

## Discussion

To evaluate the impact of presenting stimuli with negative and positive emotional content, along with emotionally neutral (humans engaged in non-emotional activities) and control stimuli (inanimate objects), on early and late attentional patterns, we designed a free viewing paradigm while recording participants' eye movements. Our results align with our hypotheses.

Firstly, we observed a significantly higher allocation of both early and late attention towards emotional stimuli (negative and positive) compared to non-emotional stimuli (neutral and control). Secondly, we found a significant early attentional bias towards negative stimuli compared to positive, neutral and control stimuli, supporting the hypothesis of a negativity bias. Thirdly, contrary to the expected positivity bias for emotional regulation, late attention predominantly remained on

**Table 2. Overall results late attention.**

| Late attention | TDF | | TNF | |
|---|---|---|---|---|
| | $\chi^2$ | *p* | $\chi^2$ | *p* |
| (Intercept) | 213.24 | **<.001** | 2543.99 | **<.001** |
| Gender | .77 | .38 | 2.01 | .16 |
| Stimulus content | 200.24 | **<.001** | 207.02 | **<.001** |
| Stimulus content * Gender | .84 | .84 | 3.6 | .31 |

*Note*. Effects of Stimulus Content (Negative, Positive, Neutral, and Control) and Participant Gender (Women, Men) on Total Duration of Fixations (TDF) and Total Number of Fixations (TNF), analyzed using generalized linear mixed models (GLMMs) with likelihood ratio tests ($\chi^2$). TDF model: conditional $R^2 = -$ (could not be estimated due to negligible variance of the random effects), marginal $R^2 = .104$. TNF model: conditional $R^2 = .31$, marginal $R^2 = .08$. Significant effects are in **bold**.

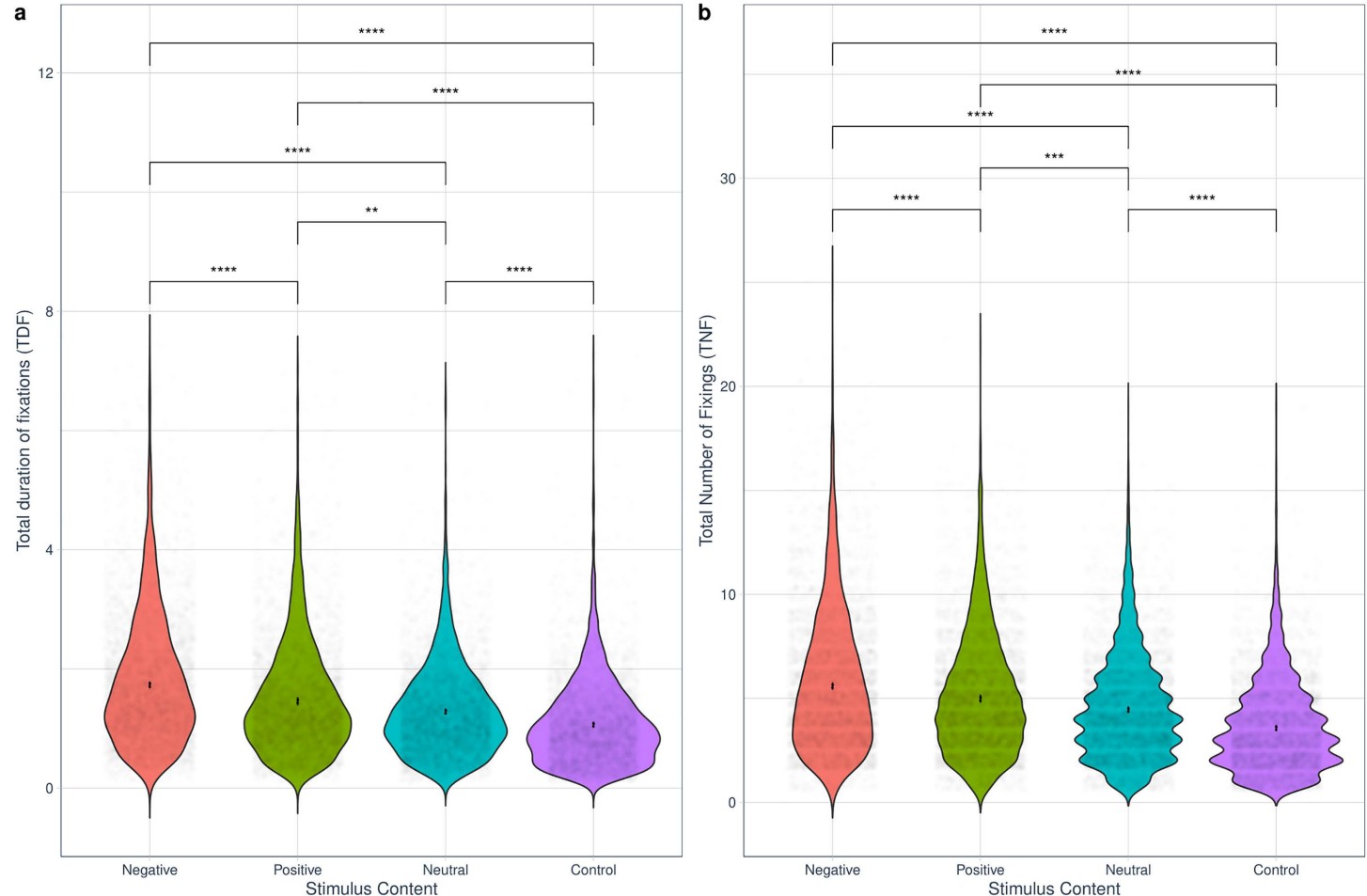

**Fig 3. Late attention.** Effects of stimuli's emotional content (negative, positive, neutral, control) on Total Duration of Fixations (TDF) and Total Number of Fixations (TNF). Significant effects are represented with lines and stars: *$p < .05$, **$p < .01$, ***$p < .001$, ****$p < .0001$.

negative stimuli compared to positive, neutral, and control stimuli. This sustained focus on negative stimuli suggests that the negativity bias may serve a regulatory function in terms of stress avoidance and the anticipation of potential threats. Lastly, consistent with our expectations, we observed increased early and late attentional patterns towards emotionally neutral stimuli containing humans compared to control stimuli represented by inanimate objects. These findings are further discussed below.

Furthermore, the models produced low marginal $R^2$ values, indicating that the fixed effects (stimulus content and gender) accounted for only a small portion of the total variance. While at first this might suggest weak explanatory power, these findings should not be overlooked. The low marginal $R^2$ values likely reflect the inherent complexity of attentional behavior, which is probably influenced by numerous unmeasured individual, contextual, and situational factors. Importantly, the detection of statistically significant effects, even with low marginal $R^2$, supports the consistency of the emotional stimuli's influence across participants. The much higher conditional $R^2$ values, which incorporate variance explained by random effects, further highlight the substantial contribution of individual- and stimulus-level variability in attentional patterns. Importantly, the presence of statistically significant effects, even with low marginal $R^2$, supports the robustness of our design and the consistency of the emotional stimuli's influence across participants.

## Early attention

Various studies have found that stimuli with emotional content elicit a faster emotional response compared to stimuli without emotional content [16,18,19,22]. Our findings align with this, as we observed a significantly faster visual response first towards negative stimuli, followed by positive stimuli and with a slower speed follow neutral stimuli and finally control stimuli, in Time to First Fixation (TFF) and First Fixation Count (FFC). This consistency supports the notion that emotional stimuli serve as powerful cues for individuals to react and adapt to contextual demands [18]. Consequently, the attentional system appears to be programmed to prioritize emotional stimuli over those lacking emotional content.

Likewise, within the realm of emotional stimuli, both TFF and FFC (but not FFD) revealed an early attentional bias towards negative stimuli over positive, neutral, and control stimuli. This supports the negativity bias hypothesis proposed by previous studies [20,22], suggesting that our attentional system prioritizes rapid processing of threatening stimuli as an adaptive mechanism for survival. This triggers automatic alertness, maximizing the ability to respond to potential environmental threats [10,20].

Moreover, particularly in tasks with low perceptual load, such as a paradigm involving the free viewing of competing emotional and non-emotional stimuli, negative stimuli tends to concentrate attentional resources, narrowing the attentional scope [10,26,59]. Consequently, there is a reduced allocation of resources for processing positive stimuli and, even more so, for non-emotional stimuli.

On the other hand, FFD, which indicates the initial level of interest in a stimulus, shows how long observers engage with and maintain attention or, conversely, if they quickly disengage from the stimulus [60]. This parameter revealed that both negative and positive emotional stimuli, as well as neutral stimuli representing humans, were more engaging than control stimuli featuring inanimate objects [61,62], in line with the findings in all parameters of early attention (and late attention) where differences were consistently observed between neutral and control stimuli, favoring greater attention to neutral stimuli than to controls.

FFD also showed that, in individuals without affective disorders, transitioning from one stimulus to another can be relatively easy, especially if these have content of interest to the observer, regardless of their emotional content (negative, positive, or neutral, but including humans). This finding supports the hypothesis of a negativity bias as an adaptive mechanism for initial attention orientation, possibly seeking to identify potential threats [63]. However, once the absence of real threat becomes conscious, it easily shifts toward other interesting stimuli.

The relevance of social interaction among humans is underscored by the attentional system's bias towards social cues sent by other humans [40], a phenomenon observed even on participants with social phobia [64].

Likewise, our results provide evidence of unreliability of neutral images included in paradigms to measure attentional biases, as they tend to elicit high variability in observers' responses [31]. This variability may arise from the inclusion of both of human and non-human stimuli within the neutral category, as observed in Sears et al. [31]. Thus, separating human stimuli from non-human stimuli would increase reliability in experimental paradigms. Indeed, a careful examination of stimulus manipulation is warranted to reliably distinguish between positive and neutral stimuli featuring humans; given the inherent value of human stimuli, confusion between these categories could potentially influence the effects of both.

## Late attention

For late attention, we expected higher TDF and TNF on stimuli with positive emotional content, followed by negative, neutral, and control stimuli, in line with the hypothesis of a positivity bias for emotional regulation in response to negative stimuli [43]. This would suggest that individuals seek to maintain a state of positive feedback, minimizing the effects of negative stimuli and facilitating emotional recovery [29,50]. However, our results contradict this prediction and the available related evidence [31,41,50], as participants maintained a significantly higher pattern of late attention (both in TDF and TNF) on negative stimuli compared to other stimulus categories. This finding challenges the positivity bias hypothesis, which would predict a preference for viewing stimuli with positive emotional content over negative and neutral stimuli, particularly during

later stages of information processing in non-clinical populations. Our results contrast with findings such as those of Isaacowitz et al. [29], who reported that older adults tended to allocate more attention to positive stimuli over time. However, the younger, non-clinical sample in our study, coupled with the simultaneous presentation of emotionally competing stimuli, likely heightened vigilance toward threat-related cues. Rather than supporting a generalized positivity bias, our findings underscore the conditional nature of this effect. One plausible explanation is that sustained attention to negative stimuli may serve as an adaptive function by facilitating learning and preparedness in response to potential threats, similar to the cognitive engagement elicited by suspenseful or dramatic scenes. This aligns with theoretical accounts proposing that negative information may be more salient because it signals possible danger and thus requires greater cognitive resources. Future research should consider factors such as participant age, task demands, cultural context, and emotional salience when evaluating attentional biases, as these variables may moderate the expression of positivity or negativity biases.

Several studies suggest that sustained attention to negative stimuli may reflect an adaptive mechanism rooted in the evolutionary role of the attentional system to prioritize potentially threatening information [21,65]. The amygdala, in particular, plays a key role in this process, enhancing the perceptual salience of emotionally salient stimuli and contributing to prolonged attentional engagement [66,67]. This bias toward negative information has been linked to stronger autonomic and cortical responses [68], which may explain the sustained fixation observed in our findings.

In everyday contexts, this attentional pattern may help explain why people are naturally drawn to emotionally intense or distressing news, images, or events. From an adaptive standpoint, prolonged attention to such content might support learning from aversive experiences and preparing for future threats. These natural biases could shape how individuals engage with emotionally charged information in their environment, possibly influencing emotional awareness, information-seeking behaviors, or even the salience of emotionally significant social cues [69,70].

Here, it is important to highlight the methodological differences across studies, particularly in scenarios where negative stimuli do not compete simultaneously with positive stimuli, leading to a lack of directly comparable evidence. For instance, Duque and Vásquez [41] examined two competing stimuli (emotional vs. neutral faces) and compared individuals with major depression to those without depression. In both cases, a bias towards happy faces was. However, this bias cannot be reliably confirmed without direct competition between emotional stimuli.

Most prior studies have presented stimuli in pairs involving neutrals rather than simultaneously presenting both positive and negative stimuli [16,19,28,41]. In our study, the simultaneous presentation of negative and positive stimuli alongside non-emotional stimuli could potentially not only mitigate the left gaze bias identified in pairwise stimulus presentations [71], but also allow for a more confident measurement of both early and late attentional priority to emotional stimuli based on their valence. In such a scenario, it might not be adaptive to shift attention towards a positive stimulus while the threat is still present, making it challenging to disengage attention from negative stimuli to redirect it towards positive ones [72].

In line with the above, individuals without affective disorders, such as anxiety and depression, may demonstrate greater emotional regulation capacity by confronting rather than avoiding negative content [73,74]. This approach may enable more effective processing of threatening experiences and potentially enhance learning and preparation for future challenges. As a result, they might experience less avoidance and a heightened responsiveness to negative stimuli, akin to the excitement and adrenaline rush associated with extreme sports [39] or the fascination triggered by unpleasant images, which can be reinforced in the brain as a means of learning about potential threats.

One hypothesis we propose to try to explain the findings is that emotional regulation strategies, such as cognitive reappraisal, may help individuals to reinterpret and process threatening stimuli more adaptively. Cognitive reappraisal allows individuals to engage with emotional stimuli without avoiding them, leading to a more effective response [75–77]. This strategy may promote sustained attention toward threatening stimuli when these are not perceived as real threats, such as in the case of the stimuli presented in our study. Just like how people might watch horror movies or observe situations in the street that don't directly affect them, they may remain engaged with these stimuli as a means to learn and prepare for potential threats. This process allows individuals to face and learn from simulated threats, which can be more adaptive

for future situations. While this framework provides a plausible but tentative explanation for our findings, further replication studies are needed to identify whether this effect is larger in other populations.

Piechaczek et al. [76], found evidence in this direction by studying cognitive reappraisal in a visual attention task. They examined attentional deployment towards emotionally negative stimuli versus neutral stimuli in two groups of adolescents (20 with depression and 28 healthy controls). Significant differences were observed between these groups, with adolescents without depression focusing more time on negative images than adolescents with depression. Although we only worked with a non-clinical sample, the control group of Piechaczek et al. [76] presents results that are in line with what was previously proposed that individuals without affective disorders may have a better capacity for emotional regulation towards negative stimuli.

In cases where stimuli are presented simultaneously, such as in the study by Lea et al. [50], where four faces with emotional expressions were presented, the results showed greater attention to happy faces. However, the study did not control for whether individuals had affective disorders, which could have biased the results.

The study most closely related to ours [31], in which four stimuli were presented simultaneously and it was ensured that individuals did not have an affective disorder, found greater late attention directed towards positive emotional content. However, in this study, neutral stimuli containing both human and non-human stimuli were not differentiated, and the sample was restricted to only.

## Constraints on generality

In this study, we excluded participants with affective disorders. While this may limit generalizability to clinical populations, it directly addresses a limitation present in previous studies, where the inclusion of participants with undetected affective symptoms may have introduced confounding effects in attentional patterns. For example, Nummenmaa et al. [19], Duque and Vázquez [41] and Sears et al. [31] either included clinical samples or did not control for the presence of affective symptoms, making it difficult to determine whether attentional biases reflect general population mechanisms or are specific to individuals with mood disorders. By ensuring a non-clinical sample, we are able to isolate attentional mechanisms characteristic of the general population, improving internal validity and reducing interpretation bias. This allows for a clearer understanding of baseline attentional responses to emotional and non-emotional stimuli.

Our study addresses a bias in interpretation compared to the rare previous studies that simultaneously presented emotionally negative, positive, and neutral stimuli without controlling for prior depression and anxiety traits. Interpreting the prioritization emotional stimuli by the attentional system and its temporal course becomes impossible without putting stimuli from these categories in competition with non-emotional stimuli, while also controlling for the presence of humans. Furthermore, controlling for affective disorders or traits reduces potential interpretation bias. Our study addresses these factors.

However, it is worth noting that, as Sears et al. [31], we screened participants for depression levels. However, we used a cutoff of 19 on the Beck Depression Inventory [48,49], unlike previous studies with a stricter cutoff (below 12 points). This disparity could affect the interpretation of our results.

Given that data collection took place amid the global COVID-19 pandemic's second year, depression scores tended to exceed 12 points, posing challenges to exclusion based on this criterion. We increased the cutoff point, considering pandemic-related lifestyles adjustments which could have affected patterns of eating, sleep, and activity –components evaluated by the Beck Depression Inventory used in our study– potentially inflating scores, but not necessarily indicative of depression [78]. It is particularly important to consider that our data were collected during the second year of the COVID-19 pandemic, a time in which emotional dysregulation, elevated depressive symptoms, and increased vigilance towards threat-related stimuli were widely documented in the general population [79,80]. The heightened attentional focus on negative stimuli may partly reflect these contextual stressors. Pandemic-related uncertainty and media exposure may have amplified attentional biases to threatening cues, making our findings particularly relevant to understanding attentional dynamics under prolonged stress.

 

These considerations may be relevant in explaining our results, as the prevailing depressed or anxious mood typical of the global crisis could have influenced a state of hypervigilance and sustained late attention towards negative stimuli [79,80]. This aligns with studies where mood is manipulated by inducing negative states, explained through the vigilance-maintenance bias hypothesis [81]. Replicating our study under post-pandemic conditions with depression scores below 12 could validate this hypothesis and test the generalizability our findings.

Indeed, considering a professional evaluation to exclude individuals with affective disorders during that time could have been ideal, as the typically reliable instruments for measuring affective traits and disorders were influenced by critical socio-contextual factors among the population.

However, while control groups are typically employed, studies often focus on individuals with affective disorders such as depression [46,47], anxiety [44], or eating disorders [82]. This study is one of the first to test attentional biases in individuals without affective disorders, using a paradigm of competing emotionally negative and positive stimuli, as well as non-emotional control stimuli while controlling for the presence of humans.

Furthermore, the simultaneous presentation of positive, negative, neutral, and control emotional and non-emotional content in our study offers advantages in interpreting attentional biases towards stimuli with emotional and non-emotional content compared to previous research. While most studies have focused on word and face stimuli [25,83], those exploring naturalistic-focused stimuli often involve only two stimuli in competition [16,19,41], leading to interpretational biases regarding differences in early and late attention towards emotionally specific content.

The validation of the emotional and non-emotional stimuli in our experimental paradigm was crucial for accurately classifying stimulus categories. While both the NAPS [54] and OASIS [55] image databases have undergone prior validation, these values may not have been fully represent the Colombian population. Therefore, pre-experimental validation of stimulus content enhances the validity to the study of attentional biases, which are of both basic cognitive interest and potential clinical application.

It is also important to consider the potential influence of socio-cultural and environmental factors on attentional patterns. Cultural norms surrounding emotional regulation, expression, and attentional control may modulate how individuals respond to emotional versus neutral stimuli. For instance, individuals from collectivist cultures may exhibit greater attentional disengagement from emotionally intense stimuli due to norms that discourage overt emotional expresión [84,85]. Moreover, contextual variables such as growing up in high-violence environments [86,87], or differences in urban versus rural upbringing [88], may sensitize or desensitize attentional systems to negative content. Sociocultural factors likely influenced the attentional patterns observed in our sample.

Considering this, it is possible that Colombia's history of armed conflict and widespread exposure to violence could have heightened sensitivity to threat-related cues. For instance, indirect exposure to neighborhood violence has been linked to increased avoidance behaviors in Colombian children [89]. Additionally, while Colombia is traditionally viewed as a collectivist society, Bogotá the urban setting of this study, has been associated with more individualistic orientations [90]. These contextual elements may shape emotional processing and attentional biases differently than in populations from more stable or culturally homogeneous regions. We recommend that future studies include contextual variables such as violence exposure, emotional suppression norms, and media use habits to better understand the generalizability of attentional biases across diverse populations.

Finally, a strength of our study lies in the inclusion of both men and women and its larger sample size (and statistical power) compared to previous studies [31,50].

## Prospects for future studies

For future studies, it is recommended to replicate this study with people without affective disorders outside the pandemic context and further investigate the emotional regulation hypothesis. In addition, they studies should consider replicating this paradigm with clinical populations, such as individuals diagnosed with depression or anxiety, to examine whether

the attentional patterns observed here differ significantly from those of nonclinical groups. Such comparisons could help clarify whether the attentional bias toward negative stimuli reflects a greater capacity for emotional regulation or vulnerability to affective dysregulation. Exploration of these dynamics in different populations would contribute to both theoretical refinement and practical applications in clinical settings. Lastly, future studies should test the interaction between stimulus content and internal emotional states by inducing states of anxiety to assess their effect on the experimental paradigm.

## Conclusion

This study contributes to the understanding of how the attentional system prioritizes emotional over neutral stimuli by using a naturalistic, free-viewing paradigm. Our findings revealed a sustained attentional bias toward negative stimuli during both early and late processing stages, even in a non-clinical population. These results suggest that, contrary to the positivity bias hypothesis, negative stimuli may continue to dominate attention in emotionally healthy individuals, possibly due to their relevance for adaptive vigilance and threat detection.

However, some limitations should be acknowledged. First, while participants with clinical diagnoses were excluded, the inclusion of individuals with subclinical depressive symptoms possibly influenced by the COVID-19 context could have affected attentional patterns toward threat-related stimuli. Second, while the covert consent procedure could raise ethical concerns, it followed approved ethical protocols and was implemented to minimize experimental bias without compromising participant autonomy. This approach aligns with previous ethical frameworks supporting covert eye-tracking under strict debriefing and withdrawal protocols [58].

*An important limitation was the size of our sample, which, although larger than in comparable studies and providing a stable estimate of the model, lacks a priori power analysis. Future research should include a priori power calculations to ensure adequate sensitivity and further validate the robustness of these findings.*

Future studies should include more diverse sociocultural samples and comparisons with clinical populations to enhance the generalizability of findings. It would also be valuable to explore how attentional biases toward negative stimuli are shaped by broader socio-ecological factors such as exposure to violence, media consumption, and perceived environmental safety beyond situational contexts like the pandemic. Additionally, examining the interaction between experimentally induced emotional states or real-time stressors and attentional allocation could provide meaningful insights for developing interventions in emotional regulation and mental health.

In addition, we present evidence for the importance of differentiating between emotional and non-emotional stimuli, controlling also for the presence of humans in non-emotional stimuli, highlighting the relevance of human stimuli for the attentional system compared to inanimate objects.

## Supporting information

**S1 File. Supplementary method, code and data analyses.**
(PDF)

## Author contributions

**Conceptualization:** Andrés Castellanos-Chacón, Daniela Arias-Otero, Milena Vásquez-Amézquita.

**Data curation:** Andrés Castellanos-Chacón.

**Formal analysis:** Andrés Castellanos-Chacón, Juan David Leongómez.

**Funding acquisition:** Andrés Castellanos-Chacón.

**Investigation:** Andrés Castellanos-Chacón, Juan David Leongómez, Milena Vásquez-Amézquita.

**Methodology:** Andrés Castellanos-Chacón, Juan David Leongómez, Milena Vásquez-Amézquita.

**Project administration:** Andrés Castellanos-Chacón, Daniela Arias-Otero, Valeria Uribe-Jaramillo, Milena Vásquez-Amézquita.

**Resources:** Andrés Castellanos-Chacón, Daniela Arias-Otero, Valeria Uribe-Jaramillo, Milena Vásquez-Amézquita.

**Software:** Andrés Castellanos-Chacón, Juan David Leongómez, Milena Vásquez-Amézquita.

**Supervision:** Juan David Leongómez, Milena Vásquez-Amézquita.

**Visualization:** Andrés Castellanos-Chacón, Juan David Leongómez.

**Writing – original draft:** Andrés Castellanos-Chacón, Daniela Arias-Otero, Valeria Uribe-Jaramillo.

**Writing – review & editing:** Juan David Leongómez, Milena Vásquez-Amézquita.

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
