## [Decision Letter · Decision Letter 0]

8 Jan 2025

Dear Dr. Castellanos-Chacon,

Please submit your revised manuscript by Feb 22 2025 11:59PM. If you will need more time than this to complete your revisions, please reply to this message or contact the journal office at plosone@plos.org . A rebuttal letter that responds to each point raised by the academic editor and reviewer(s). You should upload this letter as a separate file labeled 'Response to Reviewers'.A marked-up copy of your manuscript that highlights changes made to the original version. You should upload this as a separate file labeled 'Revised Manuscript with Track Changes'.An unmarked version of your revised paper without tracked changes. You should upload this as a separate file labeled 'Manuscript'.

We look forward to receiving your revised manuscript.

Kind regards,

Elisa Scerrati

Academic Editor

PLOS ONE

Journal Requirements:

Authors: Castellanos, A., Arias-Otero, D., Uribe-Jaramilo, V., Vásquez-Amézquita, M. y Leongómez, J.

Grant numbers awarded to each author: 1

The full name of each funder: Ministerio de Ciencia Tecnología e Innovación

URL of each funder website: https://minciencias.gov.co/viceministerios/talento/vocaciones/jovenes

Sponsors had no role in the study.

Reviewers' comments:

Reviewer's Responses to Questions

**Comments to the Author**

1. Is the manuscript technically sound, and do the data support the conclusions?

Reviewer #1: Yes

Reviewer #2: Partly

2. Has the statistical analysis been performed appropriately and rigorously?

Reviewer #1: Yes

Reviewer #2: No

3. Have the authors made all data underlying the findings in their manuscript fully available?

Reviewer #1: Yes

Reviewer #2: Yes

4. Is the manuscript presented in an intelligible fashion and written in standard English?

Reviewer #1: Yes

Reviewer #2: No

Reviewer #1: Abstract

- statistical analysis should be reported in the abstract.

Introduction

- The introduction is generally lengthy, and some sections may need to be summarized to allow the reader to grasp the key points more quickly.

- Some concepts, such as “bias” and “inhibitory processes,” could be defined more clearly.

Materials and Methods

- While it is mentioned that the sample size was increased based on previous studies, the lack of determination of the sample size based on statistical power could undermine the validity of the results. Further explanation in this regard could be beneficial.

- While the approval from the ethics committee is mentioned, additional details regarding the management of personal information and assurance of informed consent from participants could enhance transparency.

- While it is noted that the study was not pre-registered, further explanation regarding the reasons for this and its impact on the validity of the results could be beneficial.

Discussion

- While the results indicate that late attention was greater towards negative content than positive, a deeper interpretation of the reasons behind these findings and their potential impacts on human behavior could have enriched the discussion.

- Although the authors have mentioned social influences, examining cultural and environmental impacts on attention patterns and the interpretation of results could have enriched the discussion.

- Although the sample includes 122 participants, the absence of affective disorders may limit the results and restrict their generalizability to specific populations.

Reviewer #2: Dear Authors,

Thank you for the work.

My overall thoughts:

This study explores how emotional stimuli affect gaze patterns using eye-tracking technology. While the topic is interesting and the methodology generally aligns with the research question, there are several significant issues that need attention. The theoretical background feels repetitive and scattered, the ethical concerns regarding covert consent are problematic, and the statistical models, while well-selected, struggle with violations of key assumptions. The discussion section tends to drift into speculation without fully addressing inconsistencies in the results. These areas need clearer articulation and refinement to make the manuscript more cohesive and impactful.

Abstract:

The abstract tries to summarize the study but ends up being a bit fragmented. The key findings aren’t presented clearly, and the statistical results lack context. The gender differences and interaction effects, which seem important, are mentioned without proper emphasis. There’s also no acknowledgment of the study's limitations, especially the ethical concerns about covert consent.

Suggestion: Rewrite the abstract to make the findings more direct and highlight key statistical outcomes and their relevance. A brief nod to the limitations would also help set expectations.

Introduction:

The introduction sets the stage but feels repetitive in places and doesn’t flow smoothly. While some relevant studies are cited, their connection to the research question isn’t always clear. The rationale for using eye-tracking technology is underdeveloped, and the research gap isn’t strongly emphasized.

Suggestion: Streamline the theoretical framework to remove redundancy and sharpen the focus on the research question. Be more explicit about what makes this study different from previous work.

Methods:

Participants: The sample size is described, but there’s no formal power analysis to justify it.

Stimuli Selection: The process of selecting and validating stimuli lacks detail. It’s unclear whether cultural relevance was considered.

Ethics: The covert consent process raises serious ethical concerns. Even if this approach received approval, a stronger justification is needed in the text.

Statistical Analyses: The use of Linear Mixed Models (LMM) is appropriate, but many assumptions were violated. Bootstrap techniques were applied, but the results are inconsistently emphasized.

Suggestion:

Include a formal power analysis for sample size justification.

Explain the cultural validation of the stimuli in more detail.

Address the ethical concerns transparently and justify the use of covert consent.

Be clear about when bootstrap results are prioritized over parametric findings.

Results:

The statistical analysis is detailed, but there are a few key issues:

The assumption violations (e.g., non-normal residuals) are concerning, and while bootstrap techniques were used, the findings are not always interpreted with this in mind.

The marginal R² values are very low, suggesting the fixed effects (stimulus type, gender) explain little variance. This isn’t discussed enough.

Gender and stimulus interactions are presented but lack depth in interpretation.

Some figures are cluttered and difficult to interpret.

Suggestion:

Make it clear how bootstrap results affect the conclusions.

Discuss the low R² values and what they mean for the overall findings.

Provide more interpretation of the gender and stimulus interactions.

Simplify and clean up the visualizations for better clarity.

Discussion:

The discussion has a tendency to drift into speculation, especially when trying to explain the findings through cognitive reappraisal. Contradictory results, like those challenging the positivity bias hypothesis, are mentioned but not fully explored. The broader implications of the findings feel underdeveloped, and the potential influence of external factors, such as pandemic-related emotional states, isn’t examined in depth.

Suggestion:

Stay grounded in the data and avoid speculative explanations.

Address contradictory findings directly and provide plausible interpretations.

Reflect more deeply on how external factors might have influenced the results.

Offer clearer implications for theory and practice.

Conclusion:

The conclusion briefly summarizes the study but feels rushed and doesn’t fully tie back to the research question or address the broader relevance of the findings. There’s also no acknowledgment of the study's limitations, particularly regarding ethical concerns and statistical issues.

Suggestion:

Connect the findings back to the research objectives more explicitly.

Acknowledge the key limitations of the study, including ethical and methodological challenges.

Provide a more forward-looking perspective on future research directions.

So finally,

Overall, the manuscript has potential, but it needs significant revisions to improve clarity, address ethical concerns, and strengthen the interpretation of results. The theoretical framing should be sharper, the statistical analysis more transparent, and the discussion more focused on the data. With these adjustments, the study could make a meaningful contribution to the field.

With Best,

**Do you want your identity to be public for this peer review?** For information about this choice, including consent withdrawal, please see our Privacy Policy

Reviewer #1: **Yes:** Jaber Alizadehgoradel

Reviewer #2: **Yes:** Metin Çınaroğlu

---

## [Author Response · Author response to Decision Letter 1]

19 May 2025

We once again thank the editor and peer reviewers for their dedication, care, transparency, and support in the publication of this article. We have worked on every detail you highlighted, but we have also responded to your questions, comments, and opportunities for improvement. We believe this article is a better version thanks to each of your comments.

---

## [Decision Letter · Decision Letter 1]

16 Jul 2025

Dear Dr. Castellanos-Chacon,

Thank you for submitting your manuscript to PLOS ONE. After careful consideration, we feel that it has merit but does not fully meet PLOS ONE’s publication criteria as it currently stands. Therefore, we invite you to submit a revised version of the manuscript that addresses the points raised during the review process.

I apologize for the extended time required to reach this decision on your manuscript. While Reviewer 1 accepted the work, Reviewer 2, despite repeated invitations, never responded to our revision request. I was therefore compelled to invite a third reviewer, whom I thank for their availability and prompt review. After careful consideration of all reviews, I am requesting Major Revisions to address critical statistical concerns raised by Reviewer 3, which echo some doubts about the bootstrapping approach also mentioned by Reviewer 2. Specifically, Reviewer 3 is concerned about the appropriateness of the statistical modeling approach and the specification of random effects structures in the mixed models. Please find the detailed reviewer comments attached. These refinements are essential for proper interpretation of your results and support of your conclusions.

Please submit your revised manuscript by Aug 30 2025 11:59PM. If you will need more time than this to complete your revisions, please reply to this message or contact the journal office at plosone@plos.org . A rebuttal letter that responds to each point raised by the academic editor and reviewer(s). You should upload this letter as a separate file labeled 'Response to Reviewers'.A marked-up copy of your manuscript that highlights changes made to the original version. You should upload this as a separate file labeled 'Revised Manuscript with Track Changes'.An unmarked version of your revised paper without tracked changes. You should upload this as a separate file labeled 'Manuscript'.

We look forward to receiving your revised manuscript.

Kind regards,

Elisa Scerrati

Academic Editor

PLOS ONE

Journal Requirements:

Reviewers' comments:

Reviewer's Responses to Questions

**Comments to the Author**

Reviewer #1: All comments have been addressed

Reviewer #3: (No Response)

2. Is the manuscript technically sound, and do the data support the conclusions?

Reviewer #1: Yes

Reviewer #3: Partly

3. Has the statistical analysis been performed appropriately and rigorously?

Reviewer #1: Yes

Reviewer #3: No

4. Have the authors made all data underlying the findings in their manuscript fully available?

Reviewer #1: Yes

Reviewer #3: Yes

5. Is the manuscript presented in an intelligible fashion and written in standard English?

Reviewer #1: Yes

Reviewer #3: Yes

Reviewer #1: All comments provided have been carefully reviewed and addressed. All feedback is acceptable and has been considered for the improvement of the manuscript

Reviewer #3: I am not an expert in the specific topic of emotional processing with eye tracking, but the authors clearly explain their aims and methods, and I was able to follow the study design and rationale. My comments primarily focus on the analytical choices and modeling strategies. There are two main reasons of concerns that should be addressed before I can properly interpret the results and assess whether the conclusions are actually supported:

1. The response time and fixation duration variables appear to be highly skewed, as is typical for time-based measures. While the use of bootstrapping is a reasonable strategy for deriving robust confidence intervals, it does not address other fundamental modeling issues, particularly those related to the link function. When dependent variables are bounded (e.g., positive times, counts), equal differences across levels of a linear predictor cannot be interpreted as reflecting equal differences on an underlying scale of the ability or construct being measured. This is primarily why generalized linear models (GLMs) should be used, apart from appropriate modeling of the error term. In the present specific case, using generalized linear (mixed) models (GLMMs), likely with a Gamma distribution and a log link, would provide a much more natural and statistically sensible approach. In this context, assumption violations are not an unfortunate surprise, but it is inherent in the data structure! GLMMs are designed to handle this. The "lme4" or the "glmmTMB" packages of R could be used for this.

2. Another limitation concerns the specification of the random-effects structure in the mixed models. The manuscript correctly states that random effects for participants and stimuli were included, which is appropriate. However, a closer look at the shared R code reveals that only random intercepts were modeled, without any random slopes, even for within-subject factors such as stimulus content. This is an important omission, as random slopes for within-subject predictors (e.g., stimulus content by participant) are often necessary to properly account for the hierarchical structure and dependencies in the data. The absence of such slopes can inflate Type I error rates and lead to misestimated standard errors. Bootstrapping does not solve this problem if the underlying model specification is inadequate.

As a minor point: the authors say that they used linear mixed models (LMMs), but why do they report F-statistics instead of the chi-square when performing ANOVA-like comparisons? In both LMMs and GLMMs I would expect the test statistic to be z-score for coefficients, and chi-square for model comparisons.

I appreciated the authors' transparency in acknowledging that no a priori power analysis was conducted.

**Do you want your identity to be public for this peer review?** For information about this choice, including consent withdrawal, please see our Privacy Policy

Reviewer #1: **Yes:** Jaber Alizadehgoradel

Reviewer #3: No

---

## [Author Response · Author response to Decision Letter 2]

1 Sep 2025

Dear Dr. Elisa Scerrati,

We would like to express our deep appreciation for your editorial work and sincerely thank all reviewers for their insightful feedback. Thanks to their contributions, we believe our manuscript is now more solid, methodologically robust, and comprehensive. We hope you will now consider it ready for publication.

In this revision, we have carefully addressed the critical statistical concerns raised by Reviewer #3. Following these recommendations, we have:

• Re-specified all analyses as generalized linear mixed models (GLMMs) using a Gamma distribution with a log link, to more appropriately model the bounded and positively skewed nature of the dependent variables.

• Included random slopes for within-subject predictors (e.g., stimulus content by participant) to better capture the hierarchical data structure and avoid inflation of Type I error.

• The interpretation of the results remained very close to what it was before. The changes, although minimal, simplified the interpretation of the results. The only change was in delayed attention, as the interaction between gender and stimulus content was no longer significant. This interaction had previously lacked theoretical relevance as it did not reflect differences within the same stimulus type between genders.

• Adjusted our reporting to use likelihood ratio chi-square tests for model comparisons.

We are also grateful to Reviewer #1 for their positive evaluation and recognition of the manuscript’s strengths.

We are confident that these changes have strengthened the analytical rigor and interpretive clarity of our work. We respectfully submit our revised manuscript along with:

• A marked-up version highlighting all changes,

• A clean version of the revised manuscript, and

• A detailed point-by-point response to reviewers’ comments.

Thank you once again for your time and commitment to ensuring the quality of the review process.

Sincerely,

Andrés Castellanos-Chacón

---

## [Decision Letter · Decision Letter 2]

2 Oct 2025

Dear Dr. Castellanos-Chacon,

Thank you for submitting your manuscript to PLOS ONE. After careful consideration, we feel that it has merit but does not fully meet PLOS ONE’s publication criteria as it currently stands. Therefore, we invite you to submit a revised version of the manuscript that addresses the points raised during the review process.

We look forward to receiving your revised manuscript.

Kind regards,

Elisa Scerrati

Academic Editor

PLOS ONE

Journal Requirements:

Reviewers' comments:

Reviewer's Responses to Questions

**Comments to the Author**

Reviewer #3: All comments have been addressed

2. Is the manuscript technically sound, and do the data support the conclusions?

Reviewer #3: Partly

3. Has the statistical analysis been performed appropriately and rigorously?

Reviewer #3: Yes

4. Have the authors made all data underlying the findings in their manuscript fully available?

Reviewer #3: Yes

5. Is the manuscript presented in an intelligible fashion and written in standard English?

Reviewer #3: Yes

Reviewer #3: I appreciate the considerable effort the authors have put into improving the statistical modeling. The use of GLMMs with an appropriate link function and the inclusion of random slopes is an important step forward in addressing my previous concerns. The manuscript is now stronger methodological ground. However, there are a few points that should be improved and would benefit from clarification and revision:

- Modeling of First Fixation Counts (FFC): The choice to analyze FFC using a linear model with ANOVA is not ideal or clear, given the non-normal and bounded nature of count data. A Poisson or negative binomial GLMM would be more appropriate and would ensure consistency with the analytic strategy applied to the other dependent variables. Retaining a linear model here weakens the elsewhere improved analytic rigor;

- Reporting of random effects: Although the revised manuscript states that random slopes have been included, the results provide no information on the magnitude of random intercepts and slopes (e.g., variance components, standard deviations, or correlations between intercepts and slope). Without these details, it is difficult to evaluate whether the inclusion of random slopes substantively contributed to the model, or whether they accounted for any meaningful variance (perhaps they were a necessary try to make, but actually useless). Providing this information would make the hierarchical modeling more transparent and convincing. (I think that in R functions like parameters::model_parameters() or sjPlot::tab_model() or other similar would do it);

- Post hoc power analysis: The added justification based on post hoc power calculations contributes little to the overall evaluation of the study. Post hoc power is generally not informative, and presenting it as evidence of adequacy may be misleading. It would be preferable to simply acknowledge that the sample size is relatively large compared to similar studies, while noting the absence of an a priori calculation as a limitation.

**Do you want your identity to be public for this peer review?** For information about this choice, including consent withdrawal, please see our Privacy Policy

Reviewer #3: No

---

## [Author Response · Author response to Decision Letter 3]

12 Nov 2025

In this revised version, we have carefully addressed the concerns raised by Reviewer #3, which primarily focused on three aspects:

• The modeling of First Fixation Count (FFC)

• The reporting and interpretation of random effects

• The inclusion of post hoc power analyses.

Accordingly, we have implemented the following changes:

• Re-specified the First Fixation Count analysis using a COM-Poisson generalized linear mixed model (GLMM).

• Expanded the reporting of random effects, providing the variance components, standard deviations, and correlations for random intercepts and slopes.

• Removed the information about post hoc power analysis.

These revisions have been reflected in the Abstract, Methods (particularly the Statistical Analysis section), Results, Discussion, and Limitations. Importantly, these updates do not alter the overall interpretation of the findings but enhance the clarity and coherence of the manuscript.

---

## [Decision Letter · Decision Letter 3]

5 Jan 2026

Effects of stimulus emotional content on gaze pattern: an eye-tracking study

PONE-D-24-50575R3

Dear Dr. Castellanos-Chacon,

We’re pleased to inform you that your manuscript has been judged scientifically suitable for publication and will be formally accepted for publication once it meets all outstanding technical requirements.

Kind regards,

Elisa Scerrati

Academic Editor

PLOS One

Additional Editor Comments (optional):

The authors report an OSF link for access to their data; however, the link appears to be inaccessible. Please ensure that the OSF repository is set to "Public" and that the provided link is functional.

Reviewers' comments:

Reviewer's Responses to Questions

**Comments to the Author**

Reviewer #3: All comments have been addressed

2. Is the manuscript technically sound, and do the data support the conclusions?

Reviewer #3: Yes

3. Has the statistical analysis been performed appropriately and rigorously?

Reviewer #3: Yes

4. Have the authors made all data underlying the findings in their manuscript fully available?

Reviewer #3: No

5. Is the manuscript presented in an intelligible fashion and written in standard English?

Reviewer #3: Yes

Reviewer #3: The authors reported an OSF link for access to their data; however, I can't access it.

For the rest, they have properly addressed my previous concerns.

**Do you want your identity to be public for this peer review?** For information about this choice, including consent withdrawal, please see our Privacy Policy

Reviewer #3: No
